# Quantization of Integrable and Chaotic Three-Particle Fermi–Pasta–Ulam–Tsingou Models

**DOI:** 10.3390/e25030538

**Published:** 2023-03-21

**Authors:** Alio Issoufou Arzika, Andrea Solfanelli, Harald Schmid, Stefano Ruffo

**Affiliations:** 1LCEMR, Faculty of Science and Technology, Abdou Moumouni University, Niamey BP 10662, Niger; 2SISSA, Via Bonomea 265, 34136 Trieste, Italy; 3INFN, Sezione di Trieste, Via Valerio 2, 34127 Trieste, Italy; 4Dahlem Center for Complex Quantum Systems, Freie Universität Berlin, 14195 Berlin, Germany; 5Istituto dei Sistemi Complessi, Via Madonna del Piano 10, 50019 Sesto Fiorentino, Italy

**Keywords:** integrable systems, chaotic Hamiltonian systems, quantum chaos, eigenstate thermalization hypothesis

## Abstract

We study the transition from integrability to chaos for the three-particle Fermi–Pasta–Ulam–Tsingou (FPUT) model. We can show that both the quartic β-FPUT model (α=0) and the cubic one (β=0) are integrable by introducing an appropriate Fourier representation to express the nonlinear terms of the Hamiltonian. For generic values of α and β, the model is non-integrable and displays a mixed phase space with both chaotic and regular trajectories. In the classical case, chaos is diagnosed by the investigation of Poincaré sections. In the quantum case, the level spacing statistics in the energy basis belongs to the *Gaussian orthogonal ensemble* in the chaotic regime, and crosses over to Poissonian behavior in the quasi-integrable low-energy limit. In the chaotic part of the spectrum, two generic observables obey the *eigenstate thermalization hypothesis*.

## 1. Introduction

The Fermi–Pasta–Ulam–Tsingou (FPUT) model [1] has widely been investigated with the aim of understanding the approach to thermodynamic equilibrium in an isolated system [2,3]. Since the beginning of these studies, the relation with integrability in Hamiltonian mechanics and the Kolmogorov–Arnold–Moser theorem [4,5,6] has been stressed [7]. More recently, the “closeness” of the model to the integrable Toda lattice has carefully been examined [8].

Here, we take a different approach to integrability in the FPUT model, with the aim of discussing its quantization. We study the *N*-particle model for periodic boundary conditions and we introduce a specific Fourier transform, which allows us to conveniently write the nonlinear interaction terms among the Fourier mode variables [9]. We then consider the three-particle case N=3 and prove that it is integrable when either the cubic nonlinearity vanishes or the quartic nonlinearity does. Interestingly, the model with only quartic nonlinearity, dubbed β-FPUT model, has an additional integral of the motion besides energy and momentum. This integral, previously discovered in Ref. [10] using an analogy with celestial mechanics, can be associated to a cylindrical symmetry, which is only manifested in Fourier modes. Already for N=4, this integral is no longer present, and what remains for higher values of *N*, even in the thermodynamic limit, is the presence of invariant subspaces in the Fourier mode space [9], whose stability has carefully been examined by Chechin and coworkers (see Ref. [11] and Refs. therein). It should be here mentioned that a study of the Einstein–Brillouin–Keller (EBK) quantization of the integrable N=3 Toda lattice was undertaken by Isola, Kantz and Livi [12]. The complete (α+β)-FPUT model, with both non-zero cubic and quartic nonlinearity is non-integrable and displays strong chaotic motion in an intermediate energy range, as we show numerically by displaying Poincaré sections.

The quantization of classical chaotic systems is still an extremely active field of research, which has its roots in the discovery of the phenomenon of *dynamical localization* by Casati, Chirikov, Ford and Izrailev [13] and in the pioneering investigation of level statistics for Sinai billiards by Berry [14]. This latter topic was further investigated by Berry and Tabor [15], who characterized specifically the Poisson statistics in the integrable limit. These studies led to the famous *universality hypothesis* by Bohigas, Giannoni and Schmit [16], who first suggested that the spectra of quantum chaotic systems are in agreement with the predictions of random matrix theory (an early account of these developments can be found in Tabor’s book [17]). The analysis of spectral statistics for smooth Hamiltonians continued with the work of Seligman, Verbaarschot and Zirnbauer [18], who studied the transition from regular to chaotic phase-space and fitted the levels with Poisson or Gaussian orthogonal ensemble (GOE) statistics, respectively, for an even parity potential.

In this paper, we quantize the three-particle FPUT model in the formalism of first quantization, and discuss its transition from integrability to chaos using methods from random matrix theory. Moreover, we check the *eigenstate thermalization hypothesis* (ETH) [19,20] for two relevant observables, kinetic energy and the additional integral of the motion. This complements works about the bosonic FPUT model [21,22,23], and in particular a recent study [24] investigating the localization transition in the thermodynamic limit.

The paper is organized as follows. In Section 2, we introduce the appropriate Fourier mode variables and prove the integrability of both the α- and β- FPUT N=3 model. In Section 3, we quantize the three-particle FPUT model and study its level statistics. Section 4 is devoted to conclusions.

## 2. The FPUT Model and the Integrable Three-Particle Case

The Hamiltonian of the (α+β)-Fermi–Pasta–Ulam–Tsingou coupled oscillator model is
(1)$$\begin{array}{c} H=\sum_{i=1}^{N}\left[\frac{p_{i}^{2}}{2}+\frac{1}{2}{\left(q_{i+1}-q_{i}\right)}^{2}+\frac{\alpha }{3}{\left(q_{i+1}-q_{i}\right)}^{3}+\frac{\beta }{4}{\left(q_{i+1}-q_{i}\right)}^{4}\right]=H_{0}+H_{\alpha }+H_{\beta }, \end{array}$$
where $q_{i}$ is the relative displacements of the *i*-th oscillator from its equilibrium position and $p_{i}$ the conjugate momentum. We choose periodic boundary conditions $q_{N+1}=q_{1}$, $p_{N+1}=p_{1}$. The unperturbed Hamiltonian can be written in quadratic form
(2)$$\begin{array}{c} H_{0}=H\left(\alpha =0,\beta =0\right)=\frac{1}{2}\left[{}^{t}pp{+}^{t}qAq\right], \end{array}$$
where ${}^{t}$ denotes the transposed of a column vector and A is the NxN symmetric matrix whose elements are
(3)$$\begin{array}{c} A_{i,j}=-\delta_{i-1,j}+2\delta_{i,j}-\delta_{i+1,j}-\delta_{i,1}\delta_{j,N}-\delta_{i,N}\delta_{j,1}. \end{array}$$

Due to the periodic boundary conditions, matrix A is degenerate. Its eigenvalues are $\mu_{k}=\omega_{k}^{2}$, where
(4)$$\begin{array}{c} \omega_{k}=2\sin\left(\frac{\pi k}{N}\right),\;\;k=0,\ldots ,N-1 \end{array}$$
are the harmonic frequencies of the linear oscillator chain. The normalized eigenvectors are $u_{n}^{0}=1/\sqrt{N},$ for k=0, $u_{n}^{k}=\sqrt{2/N}\sin\left(2\pi kn/N+\gamma \right)$, for k=1,…,[(N−1)/2], $u_{n}^{N/2}={\left(-1\right)}^{n}/\sqrt{N}$, for k=N/2 (N even), $u_{n}^{N-k}=\sqrt{2/N}\sin\left(2\pi kn/N+\gamma +\pi /2\right)$, $u_{n}^{N-k}=\sqrt{2/N}\sin\left(2\pi kn/N+\gamma +\pi /2\right)$, for k=⌊N/2⌋+1,…,N−1. Here, ⌊•⌋ denotes the integer part, the index *n* runs from 1 to *N* and the free parameter γ can be fixed arbitrarily, due to the degeneracy of A. The NxN matrix S, which maps Fourier modes $Q_{0},\ldots ,Q_{N-1}$ into $q_{1},\ldots ,q_{N}$, is orthogonal, and its columns are given by the eigenvectors of A. The momenta in the two dual spaces, $P_{0},\ldots ,P_{N-1}$ and $p_{1},\ldots ,p_{N}$, are related by the same linear transformation S. Although the choice of γ does not affect the quadratic part of the Hamiltonian in Fourier modes, it does have consequences on the expression of the nonlinear interaction terms among the Fourier modes. A particularly convenient choice is γ=π/4, for which all columns of the transformation matrix S take the same expression $1/\sqrt{N}\left[\sin\left(2\pi kn/N\right)+\cos\left(2\pi kn/N\right)\right]$, with n=1,…,N as the row index and k=0,…,N−1 as the column index, respectively. Using this transformation, the nonlinear cubic term of the Hamiltonian is transformed into
(5)$$\begin{array}{c} H_{\alpha }\left(Q\right)=\frac{\alpha }{3}{\left(\frac{2}{N}\right)}^{3/2}\sum_{n=1}^{N}\sum_{k_{1},k_{2},k_{3}=0}^{N-1}U_{n,k_{1}}^{\pi /4}U_{n,k_{2}}^{\pi /4}U_{n,k_{3}}^{\pi /4}\omega_{k_{1}}\omega_{k_{2}}\omega_{k_{3}}Q_{k_{1}}Q_{k_{2}}Q_{k_{3}}\;, \end{array}$$
where
(6)$$\begin{array}{c} U_{n,k}^{\gamma }=\cos\left(\frac{\pi k(2n+1)}{N}+\gamma \right)\;. \end{array}$$

The sum over *n* in Equation (Equation 5) can be explicitly performed using the properties of the $U^{\prime }s$ (see Appendix A). The result is
(7)$$\begin{array}{c} H_{\alpha }\left(Q\right)=\frac{\alpha }{6}\left(\frac{\sqrt{2}}{N\sqrt{N}}\right)\sum_{k_{1},k_{2},k_{3}=0}^{N-1}B_{k_{1},k_{2},k_{3}}\omega_{k_{1}}\omega_{k_{2}}\omega_{k_{3}}Q_{k_{1}}Q_{k_{2}}Q_{k_{3}}\;, \end{array}$$
where
(8)$$\begin{array}{c} B_{k_{1},k_{2},k_{3}}=-\Delta_{k_{1}+k_{2}+k_{3}}+\Delta_{k_{1}+k_{2}-k_{3}}+\Delta_{k_{1}-k_{2}+k_{3}}+\Delta_{k_{1}-k_{2}-k_{3}}\;, \end{array}$$
and $\Delta_{r}={\left(-1\right)}^{m}$ if r=mN and 0 otherwise, with *m* an integer. The $B_{k_{1},k_{2},k_{3}}$ coupling coefficients incorporate momentum conservation. The same procedure can be used to derive the quartic term in the Hamiltonian, obtaining
(9)$$\begin{array}{c} H_{\beta }\left(Q\right)=\frac{\beta }{8N}\sum_{k_{1},k_{2},k_{3},k_{4}=0}^{N-1}C_{k_{1},k_{2},k_{3},k_{4}}\omega_{k_{1}}\omega_{k_{2}}\omega_{k_{3}}\omega_{k_{4}}Q_{k_{1}}Q_{k_{2}}Q_{k_{3}}Q_{k_{4}}\;, \end{array}$$
where
(10)$$\begin{array}{c} C_{k_{1},k_{2},k_{3},k_{4}}=-\Delta_{k_{1}+k_{2}+k_{3}+k_{4}}+\Delta_{k_{1}+k_{2}-k_{3}-k_{4}}+\Delta_{k_{1}-k_{2}+k_{3}-k_{4}}+\Delta_{k_{1}-k_{2}-k_{3}+k_{4}} \end{array}$$
again reflecting momentum conservation. This method is fully general and can be applied to derive the Hamiltonian in Fourier modes Q for all values of *N*.

The first nontrivial value of *N* is N=3, since the N=2 case, being a two-body problem, is totally integrable. For N=3, we obtain the following potential:(11)$$\begin{array}{c} V\left(Q_{1},Q_{2}\right)=\frac{3}{2}\left(Q_{1}^{2}+Q_{2}^{2}\right)+\frac{\alpha }{2}{\left(Q_{1}+Q_{2}\right)}^{3}+\frac{9\beta }{8}{\left(Q_{1}^{2}+Q_{2}^{2}\right)}^{2} \end{array}$$
where, for convenience, the Hamiltonian is rewritten in slightly different variables where $Q_{1}$ is swapped with $Q_{2}$ and, then, the sign of $Q_{1}$ is changed, leaving the Hamiltonian invariant. Therefore, we can represent the dynamics in Fourier space as the motion of a particle in the two-dimensional potential (Equation 11).

Let us first discuss the potential (Equation 11) with α=0 and β≠0. The kinetic term of the Hamiltonian is
(12)$$\begin{array}{c} K=\frac{1}{2}\left(P_{0}^{2}+P_{1}^{2}+P_{2}^{2}\right)\;, \end{array}$$
and hence, the center of mass motion is free, ($P_{0}=const.$). The potential is invariant under rotations in the $\left(Q_{1},Q_{2}\right)$ plane; therefore, the pseudo-angular momentum
(13)$$\begin{array}{c} L_{0}=Q_{1}P_{2}-Q_{2}P_{1} \end{array}$$
is conserved. Going back to the original coordinates with an inverse Fourier transform, it can be shown that
(14)$$\begin{array}{c} L_{0}=\frac{1}{\sqrt{3}}\left[p_{1}\left(q_{3}-q_{2}\right)+p_{2}\left(q_{1}-q_{3}\right)+p_{3}\left(q_{2}-q_{1}\right)\right] \end{array}$$
which was known to be a constant of motion, see [10]. In Fourier coordinates, this unexpected constant of motion acquires an explicit physical meaning, by associating it to a rotational symmetry. Thinking of the three coordinates $\left(q_{1},q_{2},q_{3}\right)$ as positions of a particle in three dimensions, this symmetry corresponds to a cylindrical rotational symmetry of the potential around the axis $\left(u_{1}^{0},u_{2}^{0},u_{3}^{0}\right)=\left(1,1,1\right)/\sqrt{3}$, the first eigenvector of A. This symmetry is related to the component of the angular momentum, $L_{0}$, along this axis. The two other constants of motion for this three-dimensional phase space are energy and the momentum along the axis $P_{0}=\left(p_{1}+p_{2}+p_{3}\right)/\sqrt{3}$.

The case α≠0, β=0 is also integrable. This can be shown by introducing the variables $s=Q_{1}+Q_{2}$ and $d=Q_{1}-Q_{2}$. In terms of these variables, the potential is written as $V\left(s,d\right)=\left(3/4\right)\left(s^{2}+d^{2}\right)+\left(\alpha /2\right)s^{3}$ and is therefore separable along the directions *s* and *d*. Along *d*, the motion is harmonic, while along *s*, the motion is the one of a cubic oscillator. The motion in the full space is therefore a composition of free, harmonic and cubic anharmonic motions along the three directions.

The intermediate case where both α≠0 and β≠0 is non-integrable. A typical potential surface is displayed in Figure 1: it shows two valleys of different depths. If the motion is within each valley, at low energy, it is quasi-regular and very weakly chaotic, close to integrable. On the contrary, when the motion runs across the two valleys, at higher energies, it is strongly chaotic, displaying intersections on the Poincaré plane that cover an area, see Figure 2 and Figure 3 with two different section planes $\left(Q_{1},P_{1}\right)$ and $\left(Q_{1},Q_{2}\right)$ (the numerical integration algorithm is described in Appendix B). More precisely, the potential minima have coordinates $Q_{1}^{-}=Q_{2}^{-}=-1/\left(\alpha -\sqrt{\alpha^{2}-3\beta }\right)$ and $Q_{1}=Q_{2}=0$, while the saddle point is located at $Q_{1}^{+}=Q_{2}^{+}=-1/\left(\alpha +\sqrt{\alpha^{2}-3\beta }\right)$ and the corresponding values of the potential energy are
(15)$$\begin{array}{c} V\left(Q_{1}^{\pm },Q_{2}^{\pm }\right)=\frac{4\alpha \left(\alpha \pm \sqrt{\alpha^{2}-3\beta }\right)-9\beta }{2{\left(\alpha \pm \sqrt{\alpha^{2}-3\beta }\right)}^{4}},\;V\left(0,0\right)=0\;. \end{array}$$

Two valleys exist for weak to moderate quartic interactions ($\beta <3\alpha^{2}$). The boundary value of the potential separating motions within the valleys and above them is given by $V_{b}=V_{+}$. Notice that in the case of strong quartic interactions ($\beta >3\alpha^{2}$), only one minimum occurs, and the motion is again quasi-integrable.

Summarizing, this analysis shows that the FPUT three-particle model is integrable in the two limits α→0,β≠0 and α≠0,β→0. Moreover, the model displays a strongly chaotic behavior for intermediate values and α and β and for energies that are close to a saddle in the two-dimensional phase-space of a fictitious particle, which represents the motion in Fourier space. These results call for an analysis of the quantum behavior of such a model, which is pursued in the following section.

## 3. The Quantum Three-Particle FPUT Model

In the remainder of the paper, we extend the study to a quantum mechanical version of the three-particle FPUT model. To this end, we promote classical coordinates $p_{j},\;q_{j}$ to quantum operators ${\hat{p}}_{j},\;{\hat{q}}_{j}$, which satisfy canonical commutation relations $\left[{\hat{q}}_{i},{\hat{p}}_{j}\right]=i\delta_{i,j}$ (ℏ=1). The quantum Hamiltonian of the three-particle FPUT model is then given by
(16)$$\begin{array}{c} H=\sum_{i=1}^{3}\left[\frac{{\hat{p}}_{i}^{2}}{2}+\frac{1}{2}{\left({\hat{q}}_{i+1}-{\hat{q}}_{i}\right)}^{2}+\frac{\alpha }{3}{\left({\hat{q}}_{i+1}-{\hat{q}}_{i}\right)}^{3}+\frac{\beta }{4}{\left({\hat{q}}_{i+1}-{\hat{q}}_{i}\right)}^{4}\right]\;. \end{array}$$
Analogously to the classical part, we can reduce the degrees of freedom by defining normal modes ${\hat{Q}}_{0}$, ${\hat{Q}}_{1}$ and ${\hat{Q}}_{2}$. The mode ${\hat{Q}}_{0}$, corresponding to center of mass motion, decouples from the other two, and we are thus left with the problem of obtaining the eigenvalues and eigenstates for the relevant normal modes ${\hat{Q}}_{1}$ and ${\hat{Q}}_{2}$ of the potential
(17)$$\begin{array}{c} \hat{V}=\frac{3}{2}\left({\hat{Q}}_{1}^{2}+{\hat{Q}}_{2}^{2}\right)+\frac{\alpha }{2}{\left({\hat{Q}}_{1}+{\hat{Q}}_{2}\right)}^{3}+\frac{9\beta }{8}{\left({\hat{Q}}_{1}^{2}+{\hat{Q}}_{2}^{2}\right)}^{2}\;. \end{array}$$

### 3.1. The Integrable Cases

As for the classical FPUT model, if α or β are equal to zero, the model is integrable. In particular, if α=0, as already discussed in Section 2, the model has a manifest cylindrical symmetry with respect to the $Q_{0}$ axis. Therefore, it is convenient to pass to cylindrical coordinates (r,ϕ,z) defined as
(18)$$\begin{array}{c} \left\{\begin{array}{c} Q_{1}=r\cos\phi \\ Q_{2}=r\sin\phi \\ Q_{0}=z \end{array}\right.. \end{array}$$
In terms of these new variables, the time-independent Schrödinger equation reads
(19)$$\begin{array}{c} \left[-\frac{1}{2r}\frac{\partial }{\partial r}\left(r\frac{\partial }{r\partial r}\right)-\frac{1}{2r^{2}}\frac{\partial^{2}}{\partial \phi^{2}}-\frac{1}{2}\frac{\partial^{2}}{\partial z^{2}}+\frac{3}{2}r^{2}+\frac{9}{8}\beta r^{4}\right]\psi =E\psi \;, \end{array}$$
where we introduced the wave function ψ=ψ(r,ϕ,z). Moreover, the Hamiltonian commutes with the operators corresponding to the two integrals of motion ${\hat{P}}_{0}$ and ${\hat{L}}_{0}={\hat{Q}}_{1}{\hat{P}}_{2}-{\hat{Q}}_{2}{\hat{P}}_{1}$. As a consequence, the energy levels can be labeled by three quantum numbers $E=E_{n,l_{0},p_{0}}$ corresponding to the three commuting observables $\hat{H},{\hat{P}}_{0},{\hat{L}}_{0}$, and we can search for the solutions of the Schrödinger equation having the separable form
(20)$$\begin{array}{c} \psi_{n,p_{0},l_{0}}\left(r,\phi ,z\right)=\frac{e^{ip_{0}z}}{\sqrt{L}}\frac{e^{il_{0}\phi }}{\sqrt{2\pi }}f_{n,l_{0}}\left(r\right)\;. \end{array}$$
The normalized eigenfunctions of the ${\hat{P}}_{0}$ and ${\hat{L}}_{0}$ operators factorize, and $f_{n,l_{0}}\left(r\right)$ is the radial wave function which remains to be found. *L* is the size of the system along the axial direction *z*. We notice that square modulus of the total wave function must be normalized to unity by requiring that
(21)$$\begin{array}{c} \int drd\phi dzr^{2}{\left|\psi_{n,p_{0},l_{0}}\left(r,\phi ,z\right)\right|}^{2}=\int drr^{2}{\left|f_{n,l_{0}}\left(r\right)\right|}^{2}=1\;. \end{array}$$
It is then convenient to introduce the new radial function $u_{n,l_{0}}\left(r\right)=\sqrt{r}f_{n,l_{0}}\left(r\right)$. Finally, inserting this ansatz in Equation (Equation 19), we obtain the radial equation
(22)$$\begin{array}{c} -\frac{1}{2}\frac{d^{2}u_{n,l_{0}}\left(r\right)}{dr^{2}}+V_{\mathrm{eff}}\left(r\right)u_{n,l_{0}}\left(r\right)=E_{n,l_{0}}u_{n,l_{0}}\left(r\right)\;, \end{array}$$
where $E_{n,l_{0}}=E_{n,l_{0},p_{0}}-p_{0}^{2}/2$, and we introduced the effective potential
(23)$$\begin{array}{c} V_{\mathrm{eff}}\left(r\right)=\frac{4l_{0}^{2}-1}{8r^{2}}+\frac{3}{2}r^{2}+\frac{9}{8}\beta r^{4}\;, \end{array}$$
which, in addition to the interaction potential $V\left(r\right)=\left(3/2\right)r^{2}+\left(9/8\right)\beta r^{4}$, also contains a centrifugal barrier term. Summarizing, we mapped the problem into a single particle moving in a one-dimensional potential. We are not aware of the exact analytical solutions of Equation (Equation 22) with given boundary conditions, but the equation could be of course solved numerically to determine the radial spectrum quantitatively. This concludes the proof of the integrability of the quantum version of the three-particle β-FPUT model.

The quantum integrability of the α-FPUT model is even simpler because the corresponding Schrödinger equation is separable in two independent one-dimensional Schrödinger equations, describing a harmonic and an anharmonic oscillator, respectively.

### 3.2. The Non-Integrable (α + β)-FPUT Three-Particle Model

The rotational symmetry of potential (Equation 11) for α=0, leading to integrability, is lost when the general case α≠0 is considered. In the classical case, we hence found chaotic behavior, as shown by the Poincarè sections. Quantum mechanically, we should expect a situation similar to the one studied in Ref. [18], where the Poisson level spacing statistics was found in the integrable limit, while these statistics were seen to belong to the GOE universality class in the case of the mixed regular–chaotic phase space. In the quantum FPUT model, we should therefore find the GOE statistics at intermediate energies and Poisson statistics at both low and high energy.

We adopt a numerical diagonalization procedure of Hamiltonian Equation (Equation 16) [25]. We discretize the Hamiltonian on a two-dimensional square lattice with grid size ${\left[-L/2,L/2\right]}^{2}$ and $N_{e}$ grid points assuming open boundary conditions. The grid is carefully chosen such that the minima of the potential are well resolved, and the boundaries of the grid are completely dominated by the quartic term. Rescaled numerical spectra are presented in Figure 4, with ascending order of energy levels. Eigenvalues stabilize as $N_{e}$ is increased, and we checked that the uncertainty in the numerical value of each eigenvalue is small compared to the mean level spacing Δ.

The central object in our analysis is the distribution function
(24)$$\begin{array}{c} P\left(s\right)=\delta \left(s-\delta_{n}\right)\;, \end{array}$$
which is obtained from consecutive level spacings
(25)$$\begin{array}{c} \delta_{n}=\frac{E_{n+1}-E_{n}}{\Delta }\;. \end{array}$$
Here, Δ denotes the mean level spacing. The spectral statistics we present are in line with the *Gaussian orthogonal ensemble* (GOE) of random matrices with distribution function [17,26,27]
(26)$$\begin{array}{c} P_{\mathrm{GOE}}\left(s\right)=\frac{\pi }{2}se^{-\frac{\pi }{4}s^{2}}\;. \end{array}$$
This poses a clear signature of a chaotic quantum system with real, normally distributed entries of the Hamiltonian. In contrast, integrability will be associated with an absence of level repulsion and a Poissonian level distribution function
(27)$$\begin{array}{c} P_{P}\left(s\right)=e^{-s}\;. \end{array}$$
Another measure for chaos is given by the level-statistics ratio
(28)$$\begin{array}{c} \bar{r}=\frac{\min(\delta_{n},\delta_{n+1})}{\max(\delta_{n},\delta_{n+1})}\;. \end{array}$$

Figure 5a shows the level distribution function P(s) for an energy window E∈[1.9,2.2] with n=300 levels. With apparent level repulsion in place, level spacings are distributed according to the GOE prediction, indicating chaotic behavior. This is contrasted with a distribution obtained in an energy window E∈[7,9] (n=300 levels) in Figure 5b, with the Poissonian level statistic. The two discussed cases confirm the physical picture presented above, though with noticeable fluctuations due to the number of considered levels.

The level spacing statistics in different regions of the spectrum is strongly affected by the shape of the potential. We will consider representative parameter values of α and β for which the potential landscape has two non-equivalent minima separated by a barrier of height $V_{b}$, see Figure 6. At very low energies, the wave function localizes in one of the potential minima, which gives rise to harmonic levels $E_{n}≃\omega \left(n+\frac{1}{2}\right)$. At very high energies, the quartic term dominates over the cubic, effectively restoring rotational symmetry, and therefore integrability. From a semi-classical point of view, we expect most of the chaos to appear near the separatrix for levels at energies $E≳V_{b}$.

The transition from Poissonian to GOE statistics as a function of energy is traced via the level ratio $\bar{r}$, see Figure 7. For this, we sweep the energy window with *n* eigenvalues centered around an energy $\bar{E}$. At high energies ($\bar{E}\gg V_{b}$), we find a level ratio consistent with the Poisson distribution, $\bar{r}\approx 0.38$. At energies near to the barrier height but sufficiently high so that it is not strongly affected by the presence of the potential minima ($V_{b}\leq \bar{E}\leq 20V_{b}$), the level ratio peaks at the GOE value $\bar{r}\approx 0.52$, signaling a transition to a fully quantum chaotic behavior of the system. The transition between these two extremal cases occurs on the same energy scale on which rotational symmetry is broken, and hence appears visible in Figure 7. For low energies, the $\bar{r}$-value decreases again but not fully to the Poissonian value since the minima support harmonic oscillator eigenfunctions. This behavior should be compared with the one of the integrable model (α=0) shown in Figure 7b. In this case, the level ratio is never compatible with chaotic statistics independently from the energy interval considered.

Real-space wave functions support the picture of chaos in distinct regions of the spectra, see Figure 6a–c. At low energies ($E<V_{b}$), back to the picture of quasi-integrability, wave functions in the potential minimum attain the shape of two-dimensional orbitals which can be labeled by radial and angular quantum numbers, Figure 6a. In contrast, the chaotic behavior for states with intermediate energies ($V_{b}<E<20V_{b}$) is reflected in irregularly shaped wavefunctions with random direction of the nodal curves, Figure 6b. At high energy ($E\gg V_{b}$), close to the other integrable limit, wavefunctions have a structure similar to the one observed at low energy, Figure 6c.

In fact, the nodal curves strongly repel each other, and do not cross, up to our numerical resolution, see Figure 8b. This was predicted by Berry [14] and first verified for a free particle in a stadium [28].

### 3.3. Eigenstate Thermalization Hypothesis

According to the *eigenstate thermalization hypothesis* (ETH) of Deutsch [19] and Srednicki [20] (see Ref. [29] for a review), if a quantum system is chaotic, the average of a generic observable taken on the eigenstates of the energy is a *smooth* function of the energy eigenvalue itself. In connection with ETH, and focusing on the relation between quantum and classical systems (e.g., for the FPUT model), the adiabatic gauge potential is introduced to characterize suppressed heating by a slow drive [30,31,32]. We checked the ETH property, restricting to the stationary solutions of the Schrödinger equation, without any attempt to analyze the time evolution. Indeed, it is also guessed that the relaxation time to the average is controlled by the size of the system, i.e., the number of particles. Therefore, we cannot expect a fast relaxation in such a low-dimensional phase space as the one of the three-particle FPUT model. The averages on the energy eigenstates of kinetic energy and angular momentum are shown in Figure 9 and Figure 10 as a function of energy in three different regions of the spectrum. Both at low and at high energy, the kinetic energy and angular momentum are irregularly scattered in a wide region of values, and change abruptly as the energy is varied. It is only in the central chaotic region of the spectrum that they appear to fluctuate stochastically around an average. We believe that this result is a particularly transparent numerical verification of the ETH hypothesis.

## 4. Conclusions

In this paper, we investigated the classical and quantum Fermi–Pasta–Ulam–Tsingou model with three particles. Classically, the model is integrable in the two limits where the cubic or the quartic term of the Hamiltonian vanishes. In the intermediate region, the model shows a mixed phase space with both regular and chaotic trajectories, in agreement with the picture suggested by the Kolmogorov–Arnold–Moser theorem. We computed the energy spectrum and showed that, as expected from the Bohigas–Giannoni–Schmit universality hypothesis, the level spacing satisfies the *Gaussian orthogonal ensemble* statistics in the chaotic part of the spectrum. In the two integrable limits, the statistics become Poissonian. We also showed how two generic observables obey the *eigenstate thermalization hypothesis* of Deutsch and Srednicki in the chaotic part of the spectrum.

## Figures and Tables

**Figure 1 entropy-25-00538-f001:**
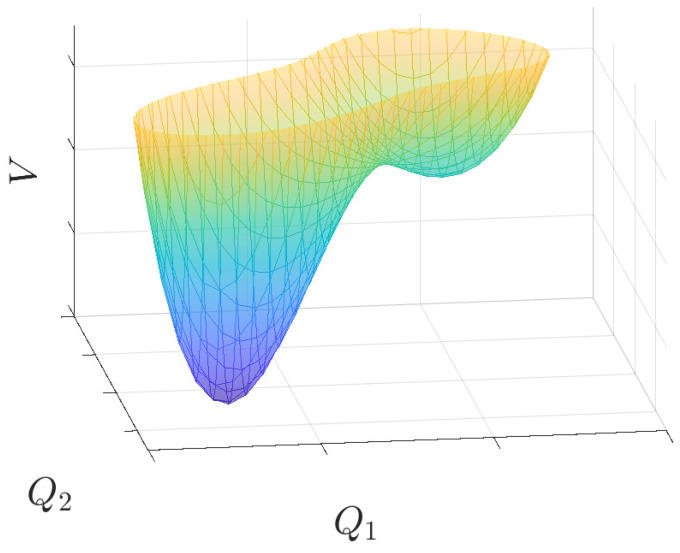
Effective potential surface $V\left(Q_{1},Q_{2}\right)=\left(3/2\right)\left(Q_{1}^{2}+Q_{2}^{2}\right)+\left(\alpha /2\right){\left(Q_{1}+Q_{2}\right)}^{3}+\left(9\beta /8\right){\left(Q_{1}^{2}+Q_{2}^{2}\right)}^{2}$ for $\alpha =\sqrt{2}$ and β=0.5.

**Figure 2 entropy-25-00538-f002:**
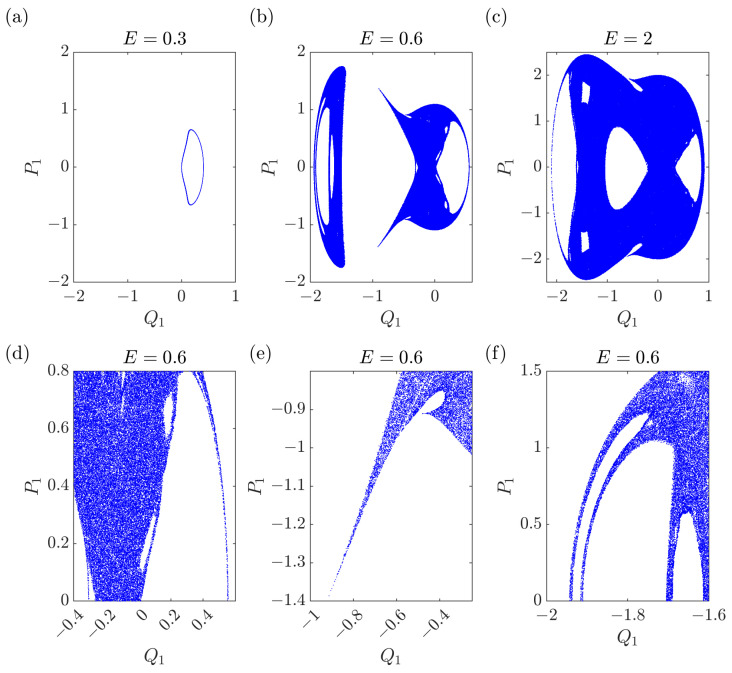
Poincaré maps $\left(Q_{1}\left(t^{*}\right),P_{1}\left(t^{*}\right)\right)$ for the particle in the potential (Equation 11) at fixed times $t^{*}$ defined by $P_{2}\left(t^{*}\right)=0$. The particle is initially at rest, and in the crater of the upper potential valley, i.e., $Q_{1}\left(0\right)>0,Q_{2}\left(0\right)=0$. (**a**) At low energies, the particle does not escape from the valley. The motion describes a closed curve and it is hence quasi-integrable. (**b**,**c**) At higher energies, the particle visits both potential valleys and a single orbit fills an area in the Poincaré section: the motion shows strong chaotic behavior. (**d**–**f**) Zooms of (**b**). Parameters: $\alpha =\sqrt{2}$ and β=0.5 and 250,000 points in the Poincare sections.

**Figure 3 entropy-25-00538-f003:**
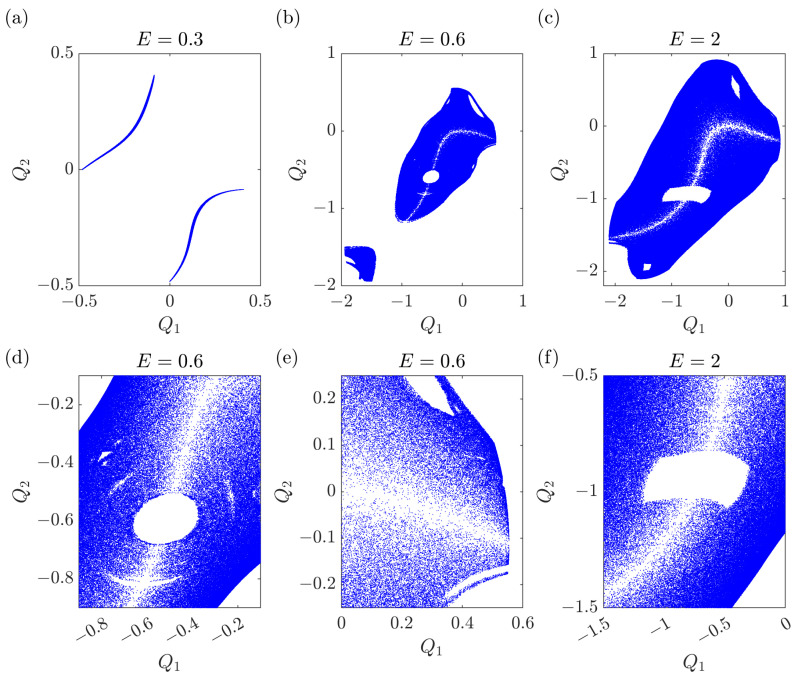
Poincaré maps $\left(Q_{1}\left(t^{*}\right),Q_{2}\left(t^{*}\right)\right)$ for fixed times $t^{*}$ defined by $P_{2}\left(t^{*}\right)=0$. Parameters are the same as in Figure 2. (**a**) Integrable regime. (**b**,**c**) Chaotic regime. (**d**,**e**) Zooms of (**b**). (**f**) Zoom of (**c**).

**Figure 4 entropy-25-00538-f004:**
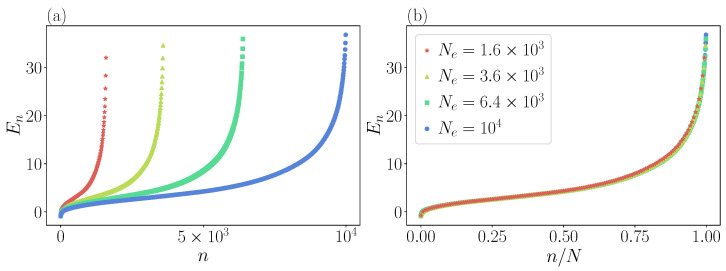
Energy levels of the three-particle FPUT model for $\alpha =\sqrt{2}$ and β=1/2. In panel (**a**), the levels are ordered in increasing value and for different grid points $N_{e}$ of the lattice. In panel (**b**), we scale the level order by the number of grid points, obtaining a perfect data collapse.

**Figure 5 entropy-25-00538-f005:**
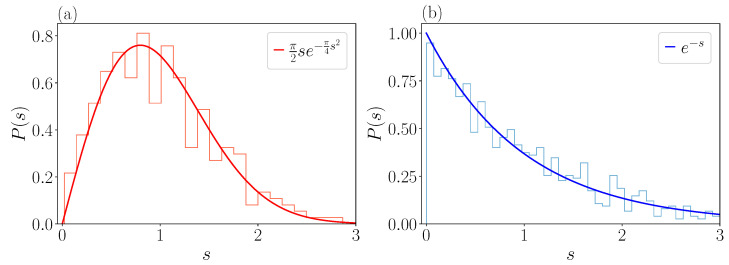
Level spacing distribution of the energy spectrum of the FPUT model P(s) with $\alpha =\sqrt{2}$, β=1/2. In panel (**a**) the distribution is well fitted with GOE statistics, while in panel (**b**) the statistics is clearly Poissonian.

**Figure 6 entropy-25-00538-f006:**
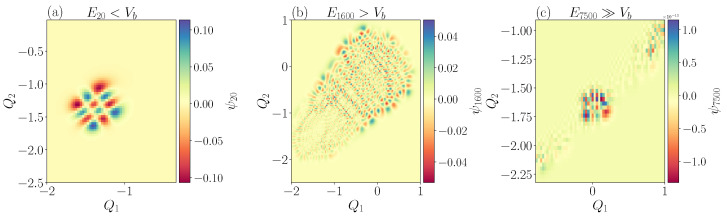
Representative wavefunctions at low $E_{20}≃-0.43<V_{b}$ (panel (**a**)), $V_{b}<E_{1600}≃2.0<20V_{b}$ intermediate (panel (**b**)) and high $E_{7500}≃7.7\gg V_{b}$ (panel (**c**)) energy.

**Figure 7 entropy-25-00538-f007:**
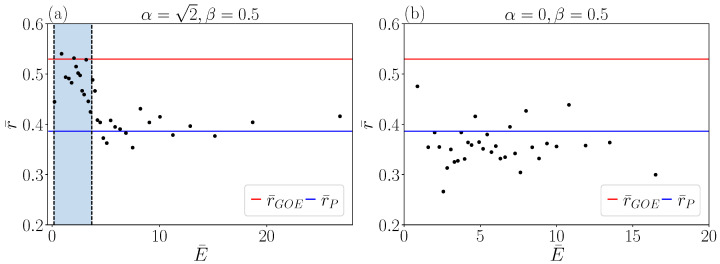
Window averaged level–statistic ratio (Equation 28) as a function of the window averaged energy for $\alpha =\sqrt{2}$, β=0.5 (panel **a**) and α=0, β=0.5 (panel **b**). The GOE and the Poisson level statistics values are shown by horizontal lines. The vertical light blue stripe indicates the region of average energy $V_{b}\leq \bar{E}\leq 20V_{b}$.

**Figure 8 entropy-25-00538-f008:**
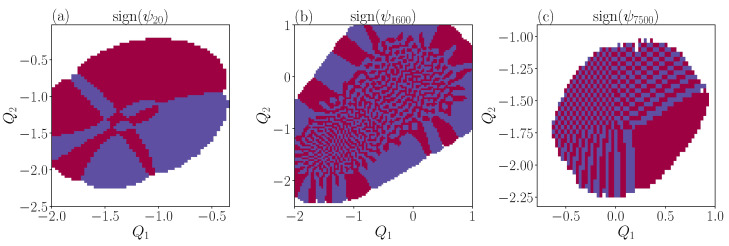
Sign of the wavefunctions, showing nodal lines in different energy regions: low energy (panel (**a**)), intermediate energy (panel (**b**)) and high energy (panel (**c**)).

**Figure 9 entropy-25-00538-f009:**
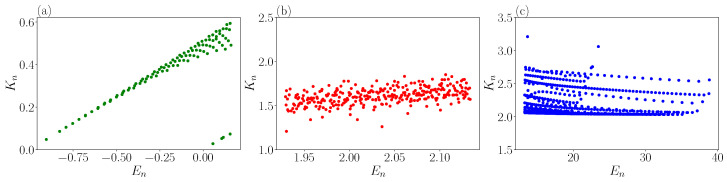
Average kinetic energy vs. energy for (**a**) the lower part of the spectrum ($E_{n}<V_{b}$), (**b**) the central part of the spectrum where eigenstates are chaotic ($E_{n}\approx 10V_{b}$), (**c**) upper part of the spectrum ($E_{n}>50V_{b}$).

**Figure 10 entropy-25-00538-f010:**
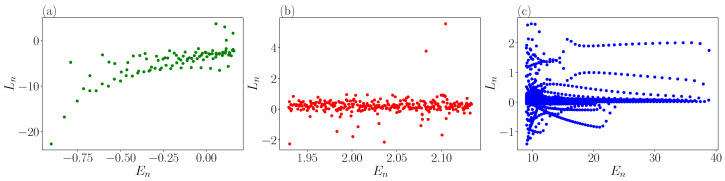
Average angular momentum vs. energy for (**a**) the lower part of the spectrum ($E_{n}<V_{b}$), (**b**) the central part of the spectrum where eigenstates are chaotic ($E_{n}\approx 10V_{b}$), (**c**) upper part of the spectrum ($E_{n}>50V_{b}$).

## Data Availability

Data available upon reasonable request.

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
