# Peer review of "Quantization of Integrable and Chaotic Three-Particle Fermi–Pasta–Ulam–Tsingou Models"

_entropy, 2023, doi:10.3390/e25030538_

Round 1

Reviewer 1 Report

The paper consider the lowest N non trivial case of the anharmonic FPUT oscillators chain, namely N=3. This is a peculiar case since when only one of the nonlinear terms is present, the model is integrable, while the generic case is not, for intermediate values of the energy. The manuscript contains analytic calculations, where the nontrivial conserved quantity is derived, while numerical simulation of Poincaré sections is employed to illustrate the intermediate, chaotic behaviour. Correspondingly, in the quantum version of the system, a transition in the level statistics is observed, as expected. In the final part of the authors provide connections to the issue of thermalization, which is a topic that recently has attracted a lot of interest.
The paper is quite interesting, the results are sound, and it fits very well to the special issue, so I suggest publication.
